# Human Hepatobiliary Organoids: Recent Advances in Drug Toxicity Verification and Drug Screening

**DOI:** 10.3390/biom14070794

**Published:** 2024-07-04

**Authors:** Haoyu Fang, Haoying Xu, Jiong Yu, Hongcui Cao, Lanjuan Li

**Affiliations:** 1Department of Pathology and Pathophysiology, Shandong Provincial Hospital Affiliated to Shandong First Medical University, Jinan 250021, China; fanghaoyu@jnl.ac.cn; 2Jinan Microecological Biomedicine Shandong Laboratory, Jinan 250117, China; yujiong@zju.edu.cn (J.Y.); ljli@zju.edu.cn (L.L.); 3State Key Laboratory for the Diagnosis and Treatment of Infectious Diseases, National Clinical Research Center for Infectious Diseases, National Medical Center for Infectious Diseases, Collaborative Innovation Center for Diagnosis and Treatment of Infectious Diseases, The First Affiliated Hospital, Zhejiang University School of Medicine, 79 Qingchun Rd., Hangzhou 310003, China; xuhaoying@zju.edu.cn; 4Zhejiang Key Laboratory for Diagnosis and Treatment of Physic-Chemical and Aging-Related Injuries, 79 Qingchun Rd., Hangzhou 310003, China

**Keywords:** hepatobiliary organoids, toxicity verification, drug screening, preclinical model, personalized medicine, human tissue-derived organoids

## Abstract

Many drug and therapeutic modalities have emerged over the past few years. However, successful commercialization is dependent on their safety and efficacy evaluations. Several preclinical models are available for drug-screening and safety evaluations, including cellular- and molecular-level models, tissue and organoid models, and animal models. Organoids are three-dimensional cell cultures derived from primary tissues or stem cells that are structurally and functionally similar to the original organs and can self-renew, and they are used to establish various disease models. Human hepatobiliary organoids have been used to study the pathogenesis of diseases, such as hepatitis, liver fibrosis, hepatocellular carcinoma, primary sclerosing cholangitis and biliary tract cancer, as they retain the physiological and histological characteristics of the liver and bile ducts. Here, we review recent research progress in validating drug toxicity, drug screening and personalized therapy for hepatobiliary-related diseases using human hepatobiliary organoid models, discuss the challenges encountered in current research and evaluate the possible solutions.

## 1. Introduction

Organoids are three-dimensional (3D) cell cultures derived from primary tissues, embryonic stem cells or pluripotent stem cells, and they are remarkably similar to their in vivo counterparts [1,2,3,4]. Over the past few decades, experimental animals, stem cells, two-dimensional (2D) cell culture models, 3D in vitro cell culture models, organoids and organ-on-a-chip models have been used in preclinical tests as a platform for testing new drugs [5]. The successful testing of new drugs is inseparable from the testing of adverse drug reactions, and adverse drug reaction syndrome is the key to completing drug-development projects and realizing drug commercialization [6]. Therefore, research on organoids plays a key role in the development and utilization of new drugs.

The liver is an essential organ with a variety of functions, such as metabolism, digestion and detoxification [7]. About 2 million people die from liver disease each year, of which 1 million die from cirrhosis complications and most others of viral hepatitis or hepatocellular carcinoma (HCC) [8]. The human biliary system consists of the intrahepatic bile duct, the extrahepatic bile duct, the cystic duct and the gallbladder [9]. Bile duct diseases are a flock of diseases characterized by a smaller bile duct, biliary blockage and bile duct hyperplasia, inflammation and fibrosis, which lead to liver damage, cholestasis and liver failure [10,11]. Hepatobiliary organoids are characteristic of the liver and have been used extensively for clinical drug testing due to their specific metabolic capabilities [12,13].

Various human tissue-derived organoids have been generated, such as colon [14,15,16], lung [17], stomach [18,19] and hepatobiliary organoids [20]. Hepatobiliary organoids were established from mouse LGR5^+^ progenitor-like elliptocytes in 2013 [21], and a successful human liver organoid was reported in 2015 [22]. Human biliary organoids were also created after liver organoids [23,24]. Clevers’ team successfully established the first tissue-derived liver organoid using bile duct fragments and Lgr5 cells sorted from mice [21]. Hepatobiliary organoids have been widely used to study fundamental processes during liver development, to model disease and identify the underlying pathological mechanisms, and to investigate regenerative medicine, drug development and precision medicine [12,25]. Disease-specific digestive organoids are used alongside healthy organoids to test drug toxicology, identify the most appropriate drug and determine the optimal dose. Here, we review recent advances in drug discovery using human hepatobiliary organoids and also elaborate on the use of hepatobiliary organoids to verify drug toxicity data (Figure 1).

## 2. Hepatic Organoids Used to Verify Toxicity

Due to the liver’s role in first-pass metabolism, detoxification and compound elimination, it can be exposed to high concentrations of drugs and their metabolites. Drugs can cause hepatotoxicity (e.g., hepatitis, liver failure, liver fibrosis and death). Drug approval requires in vivo animal studies, but animal data may have limited applicability to humans [26]. Hepatic organoid models from human cells provide a more ethically acceptable alternative and can be more easily translated. Furthermore, organotypic liver models are characterized by robust liver function and physiological-level expressions of drug-metabolizing enzymes and drug-transport proteins, facilitating a reliable prediction of hepatotoxicity [27]. The secretion of albumin in human pluripotent stem cell-derived liver organoids was observed to decrease following exposure to acetaminophen and fialuridine, indicating the occurrence of drug-induced hepatocyte injury. Notably, organoids treated with fialuridine and methotrexate exhibited significant lipid accumulation, while those treated with methotrexate displayed significant fibrosis. These findings (Table 1) indicate that liver organoids serve as effective models for liver injury and can replicate diverse toxicity responses to various drugs [28].

### 2.1. Validating Drug Toxicity in Western Medicine

Mun et al. developed a novel human pluripotent stem cell-derived hepatocyte-like liver organoid. The organoids expressed phase I drug-metabolizing cytochrome P450 (CYP) enzymes and phase II detoxification enzymes at similar levels to liver tissue transcriptome profiles. So, the researchers used them for toxicological outcome prediction. The results showed that the organoids were more sensitive to the validated hepatotoxic drugs troglitazone and trovafloxacin than the 2D hepatocyte model [29]. Shinozawa et al. successfully developed an organ-like assay to assess drug viability and cholestasis or mitochondrial toxicity by measuring hepatotoxicity pools. The results show that the method was highly predictive [30]. They developed a lipotoxicity model based on liver-like organoids and validated the model using thiazolidinediones to show that human liver organoids may highlight susceptibility to drug-induced liver injury (DILI), with positive and negative predictive power. This approach was also used to screen CYP 2C9-mediated drug-induced cholestasis variants, which showed negative and positive predictive ability, as seen in humans. Crignis et al. treated in vitro hepatitis B virus (HBV)-infected organoids and HepG2 cell (a human hepatocellular carcinoma cell line) with different concentrations of 5-fluorouracil (5-FU), using cell viability as an outcome indicator. They showed that 5-FU-induced toxicity was evident and quantifiable in the primary human liver organoid model [31]. By pairing patient-derived on-chip livers and known hepatotoxic drugs with on-chip human liver organoids (HLOs), Zhang et al. revealed numerous signs of toxicity, including increased albumin, CYP450 expression and alanine aminotransferase (ALT)/aspartic transaminase (AST) release. On-chip HLOs are supposedly capable of forecasting the synergistic hepatotoxicity of tenofovir–inarigivr [34]. 

### 2.2. Herbal Medicine Toxicity Validation

Herbal medicines are an important cause of DILI. Herbal medicines have a complex chemical composition, making drug toxicity studies difficult. Polygonum multiflorum (PM, named He Shou Wu in Chinese) has detoxifying, carbuncle-eradication, bowel-relaxing and malaria-preventive properties, but its widespread use is concerning as it causes liver damage. The bioactive components of PM include tetrahydroxy stilbene glycosides (SGs), anthraquinone, phospholipids and tannins. Among them, 2,3,5,4′-tetrahydroxy trans-stilbene-2-O-β-glucoside (trans-SG) is the main hepatotoxic component. However, Liu et al. found a higher cis-isomer content in PM samples collected from patients with liver injury [32]. Then, using a 3D cultured organoid model, they found that the increased cis-isomer-induced hepatotoxicity was similar to that observed in a rat model, with significantly increased lactate dehydrogenase (LDH), ALT and AST activity. Employing high content imaging and analysis, they further identified the lesion mechanism of the hepatotoxic cis-isomer, related to mitochondria injury. However, cell viability and related markers, such as LDH, ALT and AST, did not show any statistical difference in a HepaRG cell (a human hepatoprogenitor cell line) model following treatment with cis-SG and trans-SG, leading to no difference in hepatotoxicity between the two stereoisomers. They also used the model to screen for mitochondrial protective compounds, and the results were consistent with reduced ALT and AST levels in patients’ serum [32]. Studies have shown that liver organoids have advantages in determining the hepatotoxicity of drugs with stereoisomers.

### 2.3. Medical Device Toxicity Validation

Di-(2-ethylhexyl) phthalate (DEHP) is a plasticizer that is widely used in polyvinyl chloride medical devices, such as syringes, infusion sets, blood transfusion sets, infusion bottles and infusion bags [33], and it is easily washed off. However, there is substantial evidence that DEPH does not covalently bind to polyvinyl chloride (PVC), resulting in massive leaching of the plasticizer from the PVC infusion system into a patient’s infusion (e.g., parenteral nutrition). Additionally, adult concentrations, reportedly, may be excessively high for infants, especially sick infants. Gaitantzi et al. used HepG2 cell line, human epithelial hepatocytes, human epithelial hepatic astrocytes and liver-like organoids co-cultured with DEHP to study cytotoxicity, intercellular interactions and metabolic enzyme expression. DEHP inhibits CYP activity in hepatocytes and induces fibrosis activation of hepatic stellate cells. The direct effect of DEHP on CYP is very consistent with the results observed in organoid models, while contradicting some results obtained in 2D cultures, deeming the organoids to be highly sensitive. These findings suggest that DEHP may cause serious adverse effects, including cholestasis and fibrosis, in young infants with immature livers when present at clinically relevant concentrations (48 μg·kg^−1^·day^−1^) [35].

These studies indicate that human liver organoids exhibit detectable toxicity to clinical drugs and medical materials in vitro and can be used to evaluate the hepatotoxicity of herbal stereoisomers. The use of human liver organoid models in hepatotoxicity research on drugs and medical materials has great prospects.

## 3. Hepatobiliary Organoids for Drug Screening

Animal models display many facets that mimic human diseases, but they are limited by the accessibility of imaging observations, the presence of confounding variables, limited availability and differences between animal and human biology. The development and application of organoids provide a new opportunity to study the pathogenesis of these diseases. Hepatobiliary organoid models can be used to summarize the life cycle of the hepatitis virus, simulate key features of the disease and identify potentially effective treatments through drug screening. Organoids are highly efficient and clinically relevant in drug development and the personalized treatment of cancer patients, and they also play a role in combination drug screening (Table 2). Therefore, hepatobiliary organoids are highly important to drug screening in major hepatobiliary-related diseases.

### 3.1. Primary Liver Cancer Drug Screening

Primary liver cancer (PLC) is classified into HCC, cholangiocarcinoma (CCA), and combined HCC/CCA (CHC) tumors [39]. Due to a lack of replicable in vitro human models that reproduce the pathophysiology of original tumors to evaluate the efficacy of candidate therapies, as well as the widespread interest in organoids as useful preclinical models, Broutier et al. removed primary cells from eight PLC patients and formed PLC organoids through long-term in vitro expansion, involving the three most common PLC 3 subtypes (HCC, CCA and CHC). Histological and subtype-specific marker analysis showed that these cancer organoids expressed diagnostic markers and retained the histological features of patient tumor tissue. To verify that the tumor organoids could recognize patient-specific drug sensitivity, they tested 29 anticancer substances, including drugs in clinical use or development, successfully screening 5 highly sensitive substances, including taselisib, gemcitabine, AZD8931, SCH772984 and dasatinib [40]. Then, Li et al. established 27 patient-derived organoid (PDO) cell lines from different liver tumor sites and tested 129 FDA-approved anticancer drugs. They concluded that the majority of the drugs were either ineffective or only effective in specific organoid lines. However, screening revealed that seven pan-active drugs showed at least moderate activity in most of these organoids, supporting the use of organoids for drug discovery and screening [61].

HCC is the most generic form of PLC and the third leading cause of cancer-related deaths worldwide [62,63]. Systemic treatment for HCC has rapidly advanced over the last decade. As of 2022, three oral multi-target agents are available, including cabozantinib, lenvatinib and sorafenib, which involve one anti-programmed cell death protein (PD)-1 (pembrolizumab), one recombinant IgG1 monoclonal antibody (ramucirumab) that specifically binds to vascular endothelial growth factor receptor 2 (VEGFR-2) and two combinations, such as the ipilimumab + nivolumab and atezolizumab + bevacizumab regimens [64]. However, these therapies only marginally prolong the overall survival of patients with HCC, and the presence of resistant tumor-initiating cells increases the likelihood of tumor recurrence [41]. Exploring a solution to this problem, Xu et al. discovered two gene target combinations that strongly inhibit HCC cell growth by utilizing combinatorial genetics en masse (CombiGEM) technique, both of which correspond to the drug inhibitor sorafenib, an N-methyl-D-aspartate receptor (NMDAR) inhibitor clinically used as a vasodilator. The researchers used a group of organoid models from different HCC patients to assess the efficacy a drug combination (ifenprodil and sorafenib) in inhibiting organoid growth and self-renewal. They showed that the drug combination, but neither drug alone, significantly reduced the number of tumor-initiating cancer cells, suggesting that ifenprodil (an NMDAR inhibitor) enhances sorafenib efficacy in the treatment of HCC [41]. Zhao et al. found that long-chain acyl-coA dehydrogenase (ACADL, a mitochondrial enzyme that catalyzes fatty acid oxidation) is frequently down-regulated in HCC, leading to poor clinical outcomes in HCC patients. Therefore, they explored the mechanism of ACADL-mediated cell growth inhibition and found that ACADL could inhibit activation of the Hippo/YAP pathway in HCC cells, thereby inhibiting the growth of HCC cells. Subsequently, a patient-derived HCC organoid model was established, and the growth of HCC-like organs with low ACADL expression was successfully inhibited using the YAP inhibitor Vitepofin. These results present targeting ACADL/YAP as a new treatment strategy for HCC [42]. In order to circumvent sorafenib resistance, Karkampouna et al. used multiple models to elucidate that CRIPTO (a small signaling protein encoded by the tumor-derived growth factor 1 gene) overexpression induces tumorigenic effects in vivo, potentially leading to sorafenib resistance. The effect of sorafenib and doxorubicin in combination with CRIPTO pathway inhibitors was tested in patient-derived xenografts (PDXs) and organoid models. Reduced culture inactivity was seen in the organoid model, suggesting that inhibiting the CRIPTO/GRP78 signaling pathway specifically enhanced the response to sorafenib, providing a way to circumvent sorafenib resistance in HCC treatment [43]. In addition, using samples from untreated HCC patients, Wang et al. discovered that sorafenib was significantly less effective against CD44-positive PDOs and that blocking Hedgehog signaling significantly decreased cell viability and increased sensitivity to sorafenib. Furthermore, when GANT61 and sorafenib were added to CD44-positive HCC cell lines and PDO, respectively, cell viability and malignant properties were inhibited with a high degree of synergy in vitro and in vivo, suggesting that the combination of a Hedgehog signaling inhibitor and sorafenib may be effective in HCC patients with high CD44 levels [44]. Another study discovered a link between sorafenib resistance development in HCC organs and the epithelial-to-mesenchymal transition (EMT)/partial EMT xerogenic cancer pathway. The mTOR inhibitor compound RTP had comparable anti-tumor activity to sorafenib in HCC organs, and it was also found that targeting the mTOR pathway could effectively treat acquired sorafenib-resistant HCC organs [65]. Bortezomib, a proteasome inhibitor, was successful in HCC preclinical studies but not in clinical trials. PDX and matching PDX-derived organoids (PDXOs) were used to determine the sensitivity of HCC PDXOs to proteasome inhibitors. The second-generation proteasome inhibitor ixazomib (Ixa) and the CDK inhibitor dinaciclib (Dina), working through the stimulation of JNK signaling, have been demonstrated to exhibit synergistic pro-apoptotic and anti-proliferative actions against HCC PDXs and PDXOs. Additionally, Ixa + Dina was superior to sorafenib in reducing the development of tumors in mice [66]. In the same way, Li et al. found that omacetaxine inhibited the growth of HCC PDOs and increased apoptosis in a large number of HCC PDO models screened for 129 anticancer drugs. The dose of omacetaxine-induced toxicity in the HCC cells was well below that in healthy human hepatocytes. These results were validated in a mouse PDX model [45]. Bortezomib was found to be a highly cytotoxic small molecule in HCC; this technique was utilized based on the outcomes of liver organoid drug screening, and its effectiveness was proven using mouse trials [67].

The malignant adenocarcinoma CCA is the second most common and deadliest primary malignancy of the liver. CCA is resistant to chemotherapy, and new therapeutic drugs are urgently needed [68]. Lampis et al. confirmed oxaliplatin and 5-FU resistance in a patient before the development of PDOs by cultivating CCA-like organoids from a patient with advanced intrahepatic cholangiocarcinoma (iCCA) and testing iCCA PDOs against a cluster of small-molecule compounds. They found that the iCCA PDOs were sensitive to AUY922 (a highly potent and selective heat shock protein 90 inhibitor), and that this sensitivity was significantly enhanced following inhibition induced by microRNA 21. These findings suggest that the heat shock protein 90 (HSP90) inhibitors could be developed to treat iCCA, and that miRNA21 could function as a sensitivity marker for these medications [46]. Different from the above experiments, Maier et al. conducted RNA sequencing and transcriptomic analyses of PDOs obtained from human CCA-like organ lines established from surgical specimens, finding that PDOs retained most of the functional features of the primary tumor transcriptome, like the MAPK pathway and PI3K-Akt signal pathway. Subsequent studies also examined the responses of PDOs and corresponding classical 2D cell lines (CCLs) to the anticancer agents commonly used in CCA (gemcitabine, doxorubicin, cisplatin and sorafenib). The results showed that gemcitabine and sorafenib had similar inhibitory effects on CCL, but the PDOs showed separate drug responses and P68 (a typical multifunctional protein best known for its double- and single-stranded RNA binding function and providing energy for bidirectional RNA double unwinding activity as ATPase) was more resistant because p68 and transcription factor modulation may be involved in cancer metastasis potential and anticancer drug resistance [47]. Hepatobiliary organoids can not only screen drugs but can also be applied to evaluate the efficacy of screened drugs. Wang et al. collected two resected intrahepatic cholangiocarcinoma (IHCC) tissue samples from two independent patients, used them to establish cancer organoids of IHCC tissue and evaluated them to determine the effectiveness of iCCA translational therapy. They used the organoids to screen the drugs 5-FU, gemcitabine, ivosidenib, paclitaxel, infigratinib and cisplatin; gemcitabine and paclitaxel exhibited the strongest inhibition of the carcinogen/carcinoid, in agreement with their efficacy in the patients from which the organoids derived [48].

### 3.2. Hepatoblastoma Drug Screening

More than 90% of malignant liver tumors in children under 5 years are caused by hepatoblastoma, the most prevalent type of childhood liver cancer. Some patients with refractory hepatoblastoma are not candidates for surgical resection and do not respond to chemotherapy [69]. Aberrant Wnt/β-catenin signaling is central to the pathogenesis of hepatoblastoma. Targeting this key signaling pathway, Saltsman et al. successfully collected tumor tissue samples from three hepatoblastoma patients and established six human liver organoids. Abnormal Wnt/β-catenin signals found in organoid models were indistinguishable from tumor tissue, suggesting that the hepatoblastoma organoids were successfully established. Subsequently, they conducted a proof-of-concept drug screen of cisplatin and 12 candidate compounds using one of the patient-derived cell lines and normal organoids. The results showed variable and often non-selective drug efficacy against tumor organoids versus normal organoids. Most drugs did not show a preferential targeting of tumor organoids, and some actually appeared more toxic to normal organoids. However, there were exceptions, such as JQ1, which was screened and shown to increase the destruction of a tumor-like organ [51].

### 3.3. Fibrolamellar Carcinoma Drug Screening

Fibrolamellar carcinoma (FLC) is a rare liver cancer, with a low reported overall 5-year survival rate, occurring most frequently in adolescents and young adults. Most patients are diagnosed with advanced disease, and surgery is often only palliative [70]. FLC is characterized by the destruction of protein kinase A (PKA) signal transduction ecology, and DNAJB1-PRKACA fusion is another important driving factor of FLC. To find new treatments for advanced FLC, Narayan NJC et al. developed a total of 21 distinct organoid lines from nine patients. DNAJB1-PRKACA fusions transcripts could be detected in the FLC organoids. Furthermore, the FLC organoids summarized the parental tumors’ histological morphology, immunohistochemistry and transcriptome. In some FLC organoids, the level of wild-type PRKACA transcripts was significantly higher than of DNAJB1-PRKACA fusion transcripts. Multiple metastatic organoids from the same patient may be contaminated with normal liver cells or normal lung cells. Organoids of primary liver tumor origin are usually more susceptible to an overgrowth of normal cells. The researchers performed high-throughput drug screening of FLC organoids in combination with a normal 2D lung cell line (MRC5) and two 2D cancer cell lines (HepG2 and human neuroblastoma cell line SK-N-SH) using finasteride, a 5-α reductase type 2 inhibitor and methotrexate in the organoids, providing a new direction for the treatment of FLC [70].

### 3.4. Metastatic Hepatic Carcinoma Drug Screening

Metastases from the bowel/colon to the liver are common in patients with colorectal cancer (CRC). Death due to metastatic colon cancer typically results from cancer metastasizing to other organs rather than the primary tumor. At the time of presentation, 20% of patients have distant metastases, and the liver is typically the first organ to experience hematogenous spread; these cases have a dismal prognosis [71]. Skardal et al. created liver-based cellular organoids inoculated with colon cancer cells to generate liver tumor-derived organoid liver metastasis models for use in vitro. They tested the liver tumor organoid system with BIO and XAV939 in combination with 5-FU, a common colon cancer drug, as a model for drug candidate screening. The results showed that the agonist (BIO) manipulated the Wnt pathway to inhibit cell proliferation while the antagonist (XAV939) allowed cell proliferation, implicating the Wnt pathway in the proliferation of tumor organoids [52]. Although many studies have revealed the intrinsic tumor heterogeneity of screened drugs, in-depth understanding of the specific level of pharmacological heterogeneity in patients with metastatic tumors in vivo is still lacking. To explore changes in in vivo drug sensitivity and in vitro gene expression relative to patients, Bruun et al. used PDOs of liver metastases from CRC to conduct in vitro pharmacogenomic analysis, drug susceptibility tests and gene expression profile analyses. The results showed that no intra-patient metastatic pharmacological heterogeneity or transcriptome-level drug sensitivity changes were found, that is, the PDOs could predict in vitro clinical drug responses [53]. Based on this result, Kryeziu et al. conducted a drug screen and functional analysis of 33 investigational anticancer drugs for liver metastases in source-class organs from patients with rectal cancer, revealing LCL161 (an inhibitor of an apoptosis-inhibiting protein) as a possible experimental treatment for metastatic rectal cancer [54]. The second-line treatment for chemotherapy-resistant CRC liver metastases with acquired KRAS mutations and elevated AURKA/c-MYC expression is EGFR pathway blockade combined with AURKA inhibition using CRC-like organoids and metastatic CRC-like organoids [55]. Primeval and metastatic tumor organs from patients were utilized to accurately screen and administer drugs to clinical patients, and the findings were matched to the patients’ clinical responses. Additionally, the majority of individuals treated with the medications previously verified in the organoids exhibited successful control of their sickness [72,73,74], successfully demonstrating the advantages of the liver tumor organoid liver metastasis model for drug screening.

### 3.5. Extrahepatic Biliary Tract Cancer Drug Screening

Gallbladder cancer (GBC) and extrahepatic cholangiocarcinoma (eCCA) are two types of extrahepatic biliary tract cancer (eBTC) and are relatively rare malignancies of the digestive system. eCCA occurs at a similar rate to intrahepatic bile duct cancer [75]. The best treatment for eBTC is surgical resection. However, only 35–68% of patients with eCCA and less than 50% of patients with GBC are candidates for surgical resection [76,77,78]. Most patients are treated with conventional chemotherapy or targeted therapy, but resistance and relapse are widespread and there is no evidence indicating that these treatments improve survival or prognosis [79]. Novel methods for detecting eBTC drug sensitivity in vitro have been documented, and considering the potential value of cancer PDOs in eBTC treatment and drug screening, Wang et al. successfully constructed a PDO for GBC and eCCA. Whole-exome sequencing showed that 75% of the organoids were highly similar to the original specimen, and these analogs were used to screen against gemcitabine, infigratinib, 5-FU, ivosidenib, paclitaxel and cisplatin. The results showed that gemcitabine had a moderate-to-significant inhibitory effect on cancer growth and was the most effective in treating eBTC. The drug screen results were confirmed by comparison with the patient clinical data [49]. Saito et al. demonstrated the biological similarity between primary biliary tract cancer (BTC) tissue and established organoids by successfully constructing patient-derived iCCA- and GBC-like organoids. These organoids were used to screen 339 drugs in clinical use, and 22 of the compounds significantly inhibited the iCCA organoids. Among them, amorolfine and fenticonazole (antifungal drugs) were non-toxic or less toxic to normal bile duct epithelial cells and significantly inhibited the BTC-derived organoids. These results suggest new drug candidates for treating BTC [50]. Tienderen et al. created a novel method for encapsulating patient-derived cholangiocarcinoma carcinoids in microcapsules to achieve a high degree of standardized carcinoid culture. Additionally, they employed 51 anticancer drugs for drug screening and found a high degree of heterogeneity between patients, demonstrating the promise of this method for individualized drug screening [80].

### 3.6. Hepatic Organoids for Viral Hepatitis Drug Screening

HBV infection is the main cause of HCC and chronic cirrhosis. Treatments for chronic HBV infection are limited and mainly consist of highly toxic interferon-alpha (IFNα)/polyethylene glycol therapy and broad-spectrum antiviral drugs, such as tenofovir or entecavir [81], which are administered life-long and pose a risk of associated side effects. Liver organoid culture platforms derived from pluripotent and adult stem cells support the entire HBV lifecycle and are appropriate for in vitro HBV infection modeling, drug screening and personalized medicine [36]. This is attractive to scientists focused on HBV, and in 2021, Crignis et al. generated an HBV infection model in healthy donor-derived liver-like organoids and performed drug-screening assays, using tenofovir and 5-FU to screen for potential HBV replication inhibitors [31]. Clark et al. recently discovered that the monovalent IAP antagonist LCL-161 promotes the eradication of HBV infection in primary human liver organoids without inducing death in uninfected cells, empowering the advancement of this medication class into clinical trials [82].

Hepatitis C virus (HCV) is a virus that can cause acute or chronic infection of the liver and lead to serious liver disease, including steatosis, cirrhosis and hepatocellular carcinoma. Early HCV treatment using pegylated IFN-α (PEG-IFN-α) in combination with ribavirin is only effective in about 50% of cases. Many people with HCV infection cannot be cured and become chronically infected [83]. Persistent liver damage can occur in chronically infected HCV patients even after treatment with direct antiviral drugs (DAAs). In 2023, Meyers et al. found that liver organoids derived from adult stem cells of HCV-infected people carry HCV and are able to successfully maintain HCV infection during long-term culture. It was concluded that HCV infection in liver organoids contributes to the progression of liver disease, adds a new pathogenesis for HCV infection and provides a new direction for the development of HCV drugs [84].

Hepatitis E is a form of acute hepatitis that occurs as an outbreak and as sporadic cases. The illness is usually self-limiting, similar to other hepatotropic viruses; however, in some cases, the disease progresses to acute liver failure [38]. Over the past decade, many studies have shown that ribavirin is effective in clearing hepatitis E virus (HEV) viremia [37]. However, resistance, suboptimal efficacy, poor tolerability and contraindications have limited the widespread use of ribavirin in hepatitis E treatment. Li et al. reported that niclosamide potently inhibits HEV infection by inhibiting NF-κB signaling, and this result was confirmed in human liver organoid models. That study also showed that IFNα and ribavirin binding to niclosamide produced antagonistic effects. Brequinar and homoharringtonine exhibit potent inhibitory activity against HEV and effectively target ribavirin-resistant variants harboring the G1634R mutation. Li et al. developed an innovative HEV model, utilizing 3D cultured human liver organoids, and confirmed the anti-HEV effects of ribavirin and mycophenolic acid (MPA) in these organoids. It is important to note that obtaining liver biopsies from HEV patients for organoid culture raises safety concerns, as performing such biopsies may pose risks. However, given that the potential clinical benefits outweigh the risks, there is a case for continuing to perform liver biopsies for organoid culture despite these challenges. Fetal and adult liver-derived organoids demonstrate high tolerance towards HEV replication. By establishing these models, they comprehensively elucidated the interaction between HEV and its host factors while successfully screening and validating antiviral drugs [85].

### 3.7. Liver Fibrosis Drug Screening

Liver fibrosis is a pathological condition caused by accumulation of the extracellular matrix in response to chronic liver injury, which causes decreased liver function, the loss of liver parenchymal cells and serious complications. Organoids similar to the human liver have been designed to express the most common pathogenic mutations in autosomal polycystic recessive kidney disease (ARPKD). In 2021, Guan et al. developed a design for liver fibrosis, cultured induced pluripotent stem cells in a series of media containing different growth factors to differentiate into liver organoids and then treated the liver organoids with genome editing technology to show the liver pathology of ARPKD (including extensive fibrosis and biliary tract abnormalities). Compared with normal liver organoids, the ARPKD organoids showed increased extracellular matrix and collagen 1A expressions and the decreased zonula occludens protein 1 expression. There was an increase in irregular bile ducts in the ARPKD organoids, and the orientation and direction of the cholangiocytes were destroyed. Then, the researchers showed that three PDGFR tyrosine kinase inhibitors (imatinib, sunitinib and crenolanib) exhibited anti-fibrotic effects on the ARPKD-like organoids [56], proving that the PDGFRB pathway contributes to the formation of fibrosis in ARPKD-like organoids. These findings demonstrate the potential of ARPKD organoids as a preclinical model for testing anti-fibrotic therapies. Transforming growth factor beta (TGFβ) was administered to human liver organoids to construct a fibrosis model, followed by high-throughput screening of four compounds (SD208, imatinib, cilofexor, silymarin) to determine their efficacy in inhibiting TGFβ-induced fibrosis. Among these compounds, SD208 was identified as a significant inhibitor of fibrosis, as induced by treatment with TGFβ, MTX or LPS [28].

### 3.8. Obesity-Related Non-Alcoholic Fatty Liver Disease Drug Screening

Obesity-related non-alcoholic fatty liver disease (NAFLD), which can advance to cirrhosis and cancer, is the most prevalent chronic liver disease in the Western world [57]. The only therapies accessible to NAFLD patients are lifestyle changes, including diet and exercise [86]. At present, there are no suitable drugs to treat non-alcoholic steatohepatitis (NASH). To find a cure for NAFLD/NASH, Gwag et al. hypothesized that CD47-blocking antibodies may be a novel treatment option for NASH and used a human organoid model of NASH, which was more relevant and translational to humans, in which anti-CD47 antibody therapy inhibited the development of organoid fibrosis and inflammation [57]. Also intending to find a NAFLD/NASH cure, Wang et al. used intrahepatic bile duct cell-like organoids and liver-like organoids to create NAFLD models. Then, they evaluated metformin and obeticholic acid (INT747) and found that INT747 therapy appeared to have a minor impact on lipid synthesis, consistent with its mechanism of action, whereas metformin substantially but marginally prevented lipid peroxidation-induced lipid buildup [58]. Models of NAFLD steatosis caused by different triggers were created using human fetal hepatocyte organs. Drug screening and mechanism analysis demonstrated that inhibiting de novo lipogenesis (DNL) (directly or indirectly) was a frequent and effective mechanism to reducing steatosis, and the screened medicines successfully ameliorated steatosis [87].

### 3.9. Hepatic Fibrosis in Primary Sclerosing Cholangitis

Primary sclerosing cholangitis (PSC) is a chronic cholestatic disease characterized by progressive inflammation and fibrosis of the bile ducts, both inside and outside the liver, ultimately leading to multifocal bile duct stenosis. At present, there is no effective drug treatment for PSC. Liver transplantation is the only way to save a patient’s life; however, about 25% of transplant patients experience a recurrence of the disease [20]. Because of the need to capture many different disease features (such as multicellular interactions, fibrosis and renal tubule structure) and study their interactions in the physiological environment, as well as differences between species (possibly involving the composition of bile and gut microbiome), animal models can only partially replicate the human cholangiopathy phenotype. Organoids derived from human original cells have the advantage of maintaining the primitive cell phenotype [88]. Chen et al. prepared exosomes derived from human placental mesenchymal stem cells (ExO^MSC^), Mdr2^−/−^ mice and multicellular organoids established from PSC patients to further investigate their anti-fibrotic effect on PSC and its detailed mechanism. The findings suggest that Exo^MSC^ improves the oversecretion phenotype and intercellular interactions in the liver Th17 microenvironment by modulating PERK/CHOP signaling that supports multicellular organoids. The study data demonstrate that Exo^MSC^ has an anti-fibrotic effect in PSC diseases by inhibiting Th17 differentiation and improving the TH17-induced microenvironment, suggesting that Exo^MSC^ has potential therapeutic value in the treatment of liver fibrosis in PSC or TH17-related diseases [89].

### 3.10. Hyperuricemia Drug Screening

Uric acid is the end product of purine metabolism in all cells, and dietary purines are responsible for about a third of the body’s daily serum uric acid production [90]. Hyperuricemia is caused by an increase in uric acid in blood circulation and is becoming increasingly common worldwide. The liver is the main organ of purine anabolism in human body, and reducing serum uric acid levels is a key strategy. To this end, Hou et al. established a 2D culture model of high-uric acid cells using human normal hepatocytes (LO2 cell line) and human normal renal tubular epithelial cell (HK 2 cell line). Although this type of model is simpler, it fails to replicate the intricate purine metabolism observed in 3D human tissues. In contrast, organoids closely mimic the normal composition and behavior of human physiological cells and organs. Therefore, they changed their thinking, separated primary hepatocytes from liver tissue by enzymatic hydrolysis, constructed a 3D organoid culture system that comprehensively characterized the proliferation and differentiation characteristics of organoids and created a hyperuricemia organ model with the basic characteristics of hepatocytes. Uric acid lowering peptide compounds were screened by inducing liver organoid differentiation, and allopurinol was used to verify the model. By verifying the uric acid lowering activities of these compounds, they proved that carnosine, anserine and puerarin have certain uric acid lowering effects in this model, reflecting the advantages of organoid models [59,60].

### 3.11. Ischemic Cholangiopathy Drug Screening

Ischemic cholangiopathy is mainly caused by bile duct injury and bile duct stenosis resulting from insufficient blood supply during liver surgery and liver transplantation. Shi et al. established human hepatogenic intrahepatic bile duct cell organoids (ICOs) to simulate ischemia and reperfusion injury in organoid cultures. They used a gas-controlled anoxic workstation that exposed ICOs first to hypoxic tension and then to normoxic conditions. Hypoxia treatments of different durations (24/48/72 h) were tested, where 72 h of hypoxia resulted in a distinct morphological change in the ICOs. It was shown that hypoxia treatment raised hypoxia-inducible factor 1 (HIF-1), TGF and N-cadherin expressions, drastically decreased organoid size and caused an apoptotic cell phenotype. Cell proliferation function returned after the organoids had been reoxygenated for 24 h. On this basis, a model for ischemic biliary tract damage was developed. Caspase inhibitors were shown to induce necrotic apoptosis during hypoxia and reoxygenation, but alpha1-antitrypsin (A1AT) exhibited a protective effect against the harm brought on by hypoxia and reoxygenation, indicating that A1AT might be employed as a biliary tract protective agent [91].

## 4. Hopes and Future Challenges

The human body has many organs, and many diseases and drugs affect more than one organ or system. Recent organoid studies have used co-culture technology and microarray to realize the functional associations between different organs and to simulate the physiological state and the functional interactions between corresponding organs [92,93,94]. The liver is an important blood storage and metabolic organ. The vascularization model of liver organoids is important for studying physiology and pathology. Although many scientists have constructed vascularized liver organ models [95], traditional liver and biliary organoids used for drug screening do not represent the physiological functions of the liver. In addition, due to a lack of the original culture behavior, dependence on the 3D medium’s quality and inability to represent the characteristics of patients’ diseases, the application of organoid research in drug screening is limited [96]. Organoid models are slower to generate, and there is no reliable and repeatable way to scale organoids compared to 2D models [97,98]. Therefore, new technologies are needed to accelerate the development of organoid production and subsequent disease modeling.

Compared with previous cell models, organoids have some advantages, including potent mitochondrial respiration, enhanced expressions of genes involved in bile acid synthesis, bile acid receptors and transporters [28], remarkable CYP450 activity [99], hepatitis model construction and support of the full replication cycle of hepatitis virus [81,83,84], and high preservation of original tumor characteristics [39,47,49,69,79]. Therefore, organoids play a crucial role in drug efficacy and safety, drug development and individualized treatment [100,101]. Personalized precision medicine based on organic compounds is feasible at the laboratory level, but only a few studies have demonstrated the reliability of the results compared with clinical data. Furthermore, the sample sizes of these studies were small and the correlations between drug efficacy/toxicity in organoid models and clinical data are weak [47,79,97]. Additionally, there was high variability among individuals, and the drug results obtained by organoid drug screening must be validated in similar, larger patient populations before clinical application. To improve the efficiency and accuracy of drug screening, combining organoids with bioengineering and organ/microfluidic technologies, which more closely mimic the in vivo microenvironment, allow the identification of toxic effects that are difficult to detect in static cell culture, as well as the toxic effects of combined drug therapy [92,102]. This is an extremely efficient method for screening suitable drugs and combination drug treatment regimens. The direct integration of liver organs with liquid chromatography–mass spectrometry (LC-MS) enables selective automatic tracking of drug metabolism. Liver organoids are equipped with a liquid chromatographic column shell that enables monitoring and analysis of drug phase I metabolism through on-line coupling of the “organ in the column” unit to LC-MS. With the application of multiplex multimodal technology, organoids are becoming an accurate and universal artificial model. Through the extensive integration of intelligent algorithms into organoid research mechanisms, the development of high-dimensional technology will improve the simulation accuracy of organoid models. In the near future, organoid co-culture may be used for larger-scale models of intercellular communication and systematic biological processes [103,104].

## 5. Conclusions

Hepatobiliary organoid models are essential for drug development and toxicity assessments. Although fully functionalizing an organoid system remains challenging, recent advances in creating hepatobiliary and related disease-specific organoids from patient-derived cells show that organoid models can advance pathogenesis research and accelerate drug discovery and personalized therapy. Cooperation with other fields will improve the similarity between hepatobiliary organs and original organs, promote new medical methods and offer the possibility of disease recovery for patients.

## Figures and Tables

**Figure 1 biomolecules-14-00794-f001:**
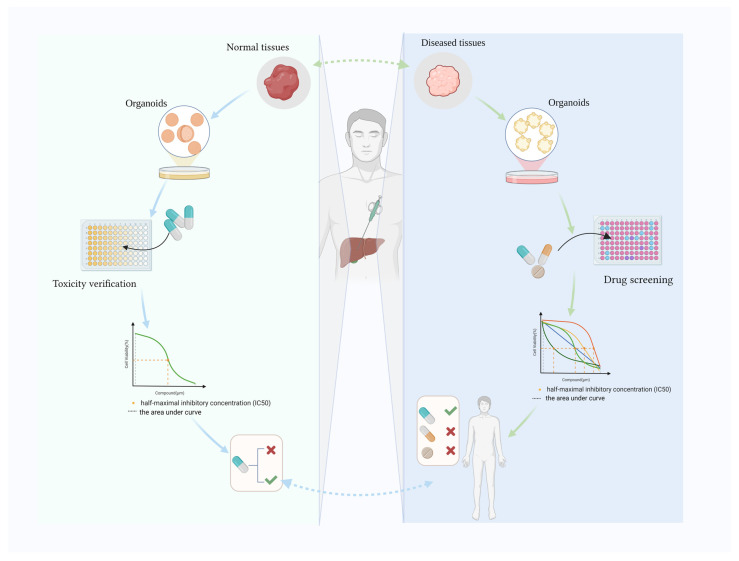
Human hepatobiliary organoids for drug toxicity verification and drug screening. Created with BioRender.com (accessed on 8 April 2024).

**Table 1 biomolecules-14-00794-t001:** Human liver organoid-based toxicity assessment.

Organoid Sources	Characteristics	Test Substances	Toxic Endpoints	Ref.
Pluripotent stem cells	Exhibit self-renewal (expandable and further able to differentiate) while maintaining their mature hepatic characteristics over long-term culture	Troglitazone, APAP, trovafloxacin and levofloxacin	Cell viability	[29]
Induced pluripotent stem cell	Contain polarized immature hepatocytes with bile canaliculi-like architecture, establishing the unidirectional bile acid-transport pathway	Hepatotoxicity library (Enzo SCREEN-WELL^®^)	Cell viability	[30]
Healthy and HBV-infected liver tissue	Support the full replication cycle of HBV, may serve as an ideal novel primary platform for drug screening as well as elucidation of the molecular events underlying HBV replication	Tenofovir and fialuridine	Relative cellular viability	[31]
HepaRG cells	Suitable to test the repeated or accumulative intoxication of drugs at low concentrations due to its long-term culturing nature	PM (trans-SG and its cis-isomer, cis-SG)	Cell viability 50, ROS, intracellular glutathione	[32]
UHCs, upcyte LSECs and UHSCs	Have typical functional characteristics of liver parenchyma including activity of cytochromes P450	Di (2-ethylhexyl) phthalate (DEHP)	mRNA expression levels, Cyp450 activities	[33]

Note: APAP, autoimmune pulmonary alveolar proteinosis; HBV, hepatitis B virus; GSH, glutathione; Ref., reference; LSECs, liver sinusoidal endothelial cells; PM, polygonum multiflorum; ROS, reactive oxygen species; trans-SG, 2,3,5,4′-tetrahydroxy trans-stilbene-2-O-β-glucoside; UHCs, upcyte human hepatocytes; UHSCs, upcyte hepatic stellate cells.

**Table 2 biomolecules-14-00794-t002:** Hepatobiliary organoids for drug screening.

Diseases	Org Type	Characteristics	Org Sources	Test Substances	Test Endpoints	Ref.
Hepatitis B	Liver org	Support the full replication cycle of HBV, may serve as an ideal novel primary platform for drug screening as well as elucidation of the molecular events underlying HBV replication	Healthy and HBV-infected liver tissue	Tenofovir and fialuridine	Relative cellular viability	[31]
		——	Liver specimens from healthy donors	The monovalent IAP antagonist LCL-161 and recombinant TNF or the combination of both	HBV DNA levels, the levels of cleaved caspase 3	[36]
Hepatitis E		Harboring subgenomic genotype 1 or3 HEV replicon	Tissue samples of donor liver biopsies	Niclosamide, ribavirin	Luciferase activity, HEV RLU	[37]
	ICOs	Support the full life cycle of HEV infection and able to anti-HEV drug discovery	Tissue samples of donor liver biopsies and fetal liver tissues	Ribavirin, mycophenolic acid, homoharringtonine and a library of 94 broad-spectrum antiviral agents	Cell viability and HEV replication–related luciferase activity	[38]
PLC	Tumor org	1. Recapitulate the histological architecture and expression profiles of the corresponding parent tumour; 2. Retain the specific differences between patients as well as between tumour subtypes	Liver tumour tissue from untreated PLC patients	29 anti-cancer compounds	Relative cellular viability, AUC and IC50	[39]
		1. Display marker profiles similar to the original primary human tumors; 2. Reflect cell-intrinsic sensitivity to drugs	Human HCC or CCA tissue	FDA-approved cancer drug library (https://wiki.nci.nih.gov/display/NCIDTPdata/Compound+Sets, accessed on 28 August 2022)	Relative cellular viability 50	[40]
HCC		Model the pathophysiology of a growing tumor in vivo, able to evaluate inhibitions of growth and self-renewal brought by the drug combination	Patient liver specimens	NMDAR antagonist ifenprodil and sorafenib	Relative cellular viability	[41]
		Display consistent expression patterns of ACADL and YAP with original tissues	Primary HCC specimens, nontumorous liver tissues	YAP inhibitor verteporfin	Relative viability of control (dose-response curves)	[42]
		——	PDX tumors	Sorafenib, N20 (GRP78 blocking peptide)	ATP content	[43]
		Maintain the histological features of the corresponding tumors and responded to drug treatment	HCC specimens	Sorafenib, Hedgehog signaling inhibitor (GANT61)	Cell viability, IC 50	[44]
		Drug screening in HCC PDOs can be utilized for drug discovery or drug repurposing	HCC specimens	129 drugs (Omacetaxin etc.)	Cell viability, IC 50	[45]
CCA		Retain the same morphology of the primary tumor, the same positivity for cytokeratin 7 and 19	CCA liver biopsies	484 small-molecule compounds	Cell viability	[46]
		The patient-derived organoids retain transcriptomic features and the mutational landscape of the parental tumor	CCA surgical resection specimens	Anti-cancer agents (gemcitabine, sorafenib, cisplatin and doxorubicin)	Relative cellular viability	[47]
IHCC		Expression profiles of cancer organoid resembled original tissue, CK7 and EpCAM were highly expressed	IHCC tissue samples	Gemcitabine, 5-FU, cisplatin, paclitaxel, infigratinib or ivosidenib, FGFR inhibitor(infigratinib)	Dose-response curves and IC50	[48]
eBTC		Different PDOs exhibited diverse growth rates during in vitro culture. Most retained the original structures of adenocarcinoma	Samples of GBC and eCCA	Gemcitabine, 5-FU, cisplatin, paclitaxel, infigratinib and ivosidenib	Cell viability, IC 50	[49]
		Harbored mutations of driver genes such as TP53 and KRAS	Cancer and non-cancer tissue specimens	A group of 339 medicines already in clinical use	Cell viability	[50]
Hepatoblastoma		Show morphologic and genomic similarity to human tissue from which they are derived	Hepatoblastoma tumor tissue and surrounding tissue	12 candidate compounds	Cell viability	[51]
Metastatic HCC	liver org	N-cadherin staining was observed in the tumor foci regions of the live-tumor organoids, co-localizing withHCT-116 cells	HepG2 hepatoma cells	5-FU	Anti-caspase 3 antibodies	[52]
		Formed clusters based primarily on the sensitivity to EGFR and/or MDM2 inhibition, as well as to SN-38 and nucleosides	Liver resection from the studied tumors	40 anticancer agents	Cell viability	[53]
		Retain the undifferentiated phenotype and expression patterns of the tumor	Samples of processed liver metastases	A custom library of 33 clinically relevant small molecule inhibitors and three 5-FU-based drug combinations with Leucovorin (FLV), Oxaliplatin (FLOX) and SN-38 (FLIRI)	Growth rate adjusted drug sensitivity scores	[54]
		KRAS/NRAS and BRAF wild-type	Partial hepatectomy normal and cancerous tissue specimens	FOLFIRI, anti-EGFR antibody cetuximab (Cmab), afatinib (pan-HER inhibitor), selumetinib (MEK inhibitor), type II inhibitor alisertib (MLN8237)	Cell viability, IC50, AUG, cell viability at Emax	[55]
Liver fibrosis		Reproduce ARPKD liver pathology, which includes biliary abnormalities and extensive fibrosis	iPSCs	The PDGFR tyrosine kinase (Crenolanib, Sunitinib and Imatinib)	Fibrosis scores	[56]
NAFLD	NASH org		Human hepatocytes, macrophages and stellate cells	Anti-CD47 antibody	Immunofluorescence staining and qPCR	[57]
	ICOs	Lipid accumulation in human liver organoids was much more robust compared to that in mouse liver organoids	Human intrahepatic bile duct tissue	Obeticholic acid (INT747) and metformin	Lipid droplet size	[58]
HUA	Liver org	As a stable model of HUA for long-term studies and screening for antihyperuricemic compounds	Human liver paracarcinomatous tissues	Allopurinol, carnosine and anserine	Concentration of uric acid	[59]
		Maintain intact purine metabolic pathways as well as mimicking the uric acid production of the human liver	Human liver paracarcinomatous tissues	Allopurinol and puerarin	Content of uric acid	[60]

Note: 5-FU, 5-fluorouracil; AUC, the area under the dose response curve; CCA, cholangiocarcinoma; eBTC, extrahepatic biliary tract cancer; eCCA, extrahepatic cholangiocarcinoma; GBC, gallbladder carcinoma; HCC, hepatocellular carcinoma; HUA, hyperuricemia; IC50, the half-maximal inhibitory concentration; IHCC, intrahepatic cholangiocarcinoma; ICOs, intrahepatic cholangiocyte organoids; NASH, nonalcoholic steatohepatitis; Org, organoid; PDX, patient-derived xenografts; PLC, primary liver cancer; Ref., reference; RLU, relative luciferase unit.

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
