# Peer review of "Human Hepatobiliary Organoids: Recent Advances in Drug Toxicity Verification and Drug Screening"

_biomolecules, 2024, doi:10.3390/biom14070794_

Round 1

Reviewer 1 Report

Comments and Suggestions for Authors

The presented manuscript is a detailed analysis of the literature on the use of liver organoids for drug screening. Such a review is of interest both to cell biologists and to representatives of pharmacology and medicine. The authors reviewed the literature on different types of organoids and drug testing for a wide range of disease entities. The structure of the review is logical, the text is easy to read. I would recommend the review for publication after minor comments have been corrected:

1. There was a typo in line 236 "becaose" instead of because. I would also recommend having the text proofread by a native speaker.

2. Section 3 stands out from the rest of the text in more difficult language. The section is summarized with an excellent figure, but it is difficult to follow the same logic when reading the text. I would recommend working on the presentation style in this section and coordinating the presentation of the material with the figure. In its current form, the text of the section looks like a list of facts and is difficult to understand. Although it contains very valuable information. I would also recommend clarifying more explicitly which experiments were performed on organoids and which on 2D cultures.

3. Section 4 reveals an extremely interesting topic about testing drugs against non-cancer pathologies. However, in my opinion, the topic of organoids itself is not sufficiently covered in this section. The section would be more complete if authors added a more detailed description of the model organoids for each pathology, indicating why this model is relevant in each case.

Comments on the Quality of English Language

I would recommend having the text proofread by a native speaker.

Reviewer 2 Report

Comments and Suggestions for Authors

The authors summariz the most recent research on the use of organoids for the toxicity evaluation of different molecules. The manuscript is fairly well writte, with minor english problems.

Some comments/suggestions:

- Abreviation list must be included to faciliate reading

- The first time that a word is abreviated must be fully written. 

- Conclusions and perspectives are only conclusions.

Other comments in the PDF file.  

Comments on the Quality of English Language
